# Cooperation of N- and C-terminal substrate transmembrane domain segments in intramembrane proteolysis by γ-secretase

Nadine T. Werner[1,7], Philipp Högel[2,7], Gökhan Güner [3,7], Walter Stelzer [2], Manfred Wozny[4], Marlene Aßfalg[3,5], Stefan F. Lichtenthaler[3,5,6✉], Harald Steiner [1,3✉] & Dieter Langosch [2✉]

Intramembrane proteases play a pivotal role in biology and medicine, but how these proteases decode cleavability of a substrate transmembrane (TM) domain remains unclear. Here, we study the role of conformational flexibility of a TM domain, as determined by deuterium/hydrogen exchange, on substrate cleavability by γ-secretase in vitro and *in cellulo*. By comparing hybrid TMDs based on the natural amyloid precursor protein TM domain and an artificial poly-Leu non-substrate, we find that substrate cleavage requires conformational flexibility within the N-terminal half of the TMD helix (TM-N). Robust cleavability also requires the C-terminal TM sequence (TM-C) containing substrate cleavage sites. Since flexibility of TM-C does not correlate with cleavage efficiency, the role of the TM-C may be defined mainly by its ability to form a cleavage-competent state near the active site, together with parts of presenilin, the enzymatic component of γ-secretase. In sum, cleavability of a γ-secretase substrate appears to depend on cooperating TM domain segments, which deepens our mechanistic understanding of intramembrane proteolysis.

[1] Biomedical Center (BMC), Division of Metabolic Biochemistry, Faculty of Medicine, LMU Munich, Munich, Germany. [2] Chair of Biopolymer Chemistry, Technical University of Munich, Freising, Germany. [3] German Center for Neurodegenerative Diseases (DZNE), Munich, Germany. [4] Am Goldhammer 11, 90491 Nürnberg, Germany. [5] Neuroproteomics, School of Medicine, Klinikum rechts der Isar, Technical University of Munich, Munich, Germany. [6] Munich Cluster for Systems Neurology (SyNergy), Munich, Germany. [7] These authors contributed equally: Nadine T. Werner, Philipp Högel, Gökhan Güner. ✉email: stefan.lichtenthaler@dzne.de; harald.steiner@med.uni-muenchen.de; langosch@tum.de

Understanding the mechanism of intramembrane proteolysis presents a formidable challenge as cleavage occurs within the plane of a lipid membrane. The aspartate protease γ-secretase cleaves the transmembrane domain (TMD) of C99, a shedded form of the amyloid precursor protein (APP) being causally linked to Alzheimer's disease. Cleavage of C99 by γ-secretase generates ~4 kDa amyloid-β (Aβ) peptides, longer forms of which are harmful and believed to trigger the disease[1,2]. In addition to C99, γ-secretase has been reported to cleave the TMDs of ~150 other proteins[3]. Having small extracellular domains is one requirement for cleavage by γ-secretase. Also, all currently known substrates share a type I, i.e., $N_{out}$, transmembrane topology, yet they represent only a small fraction of this class of single-span membrane proteins. Their TMDs do not share an apparent consensus motif[4]. Nonetheless, substrate cleavage shows site specificity which is influenced by many disease-associated and artificial point mutations[1,5].

The sequence-specificity of substrate cleavage in the absence of common sequence patterns presents a conundrum, which led to the view that the presence of certain structural features of a substrate may determine its cleavability by γ-secretase. In seminal studies, the NMR structures of C99 revealed a bend at the $G_{37}G_{38}$ (Aβ numbering) motif within TM-N, the N-terminal half of its transmembrane (TM) helix[6,7]. Considerable conformational flexibility at the bend was confirmed by deuterium/hydrogen exchange (DHX) experiments and molecular dynamics (MD) simulations[8–10]. Indeed, mutations altering helix flexibility at $G_{37}G_{38}$ affect the efficiency and specificity of cleavage[10,11]. It had therefore been suggested that substrate TM helices might share a flexible TM-N (reviewed in ref. [12–14]).

Another potential mechanism governing substrate selection has been proposed to rely on helix flexibility around the cleavage sites. In the case of soluble proteases, the part of a substrate that is docked into a protease's active site exists in an extended conformation exposing the scissile bond to the catalytic residues[15]. Accordingly, substrates of soluble proteases tend to exhibit enhanced conformational flexibility around the scissile peptide bonds. Therefore, cleavage sites are abundant in loop regions as well as in helical regions that are predicted to unfold easily[16]. That an intramembrane protease also requires partial substrate unfolding has only recently been demonstrated by the cryo-EM structures of γ-secretase in complex with the APP fragment C83 or with a fragment of Notch1, another major γ-secretase substrate. Both structures reveal helix unfolding around the residues forming the initial cleavage sites[17,18]. Similarly, interaction with a homolog of presenilin, the catalytic subunit of γ-secretase, had resulted in extended substrate conformation[19,20]. Therefore, one might expect that substrate TMD helices are highly flexible near their scissile bonds and that further mutational destabilization would promote cleavage. In line with this, γ-secretase cleavage of TREM2 has been reported recently to be located within a presumably flexible part of its TMD[21]. However, mutational studies have not established a clear link between the impact of mutations on helix flexibility near the scissile sites and the efficiency of their cleavage[22–26].

Here, we initially compared the flexibility profiles of the C99 TMD to those of other well-established γ-secretase substrates. Focusing on C99, we then employed a gain-of-function approach to systematically explore the relationship between its TM helix flexibility and its cleavability with a view to delineating the mechanism of substrate/non-substrate discrimination. To this end, we first designed a non-substrate based on a rigid TM helix. Grafting different motifs of the natural C99 TMD onto this template identified sequence motifs as being crucial for flexibility and/or cleavability. Indeed, the $V_{36}G_{37}G_{38}V_{39}$ motif within TM-N conferred partial cleavability along with pronounced helix flexibility in DHX experiments. However, flexibility within the natural cleavage region of TM-C appeared not to govern substrate selection although TM-C is required for full cleavability. Our data suggest that the cooperation of N- and C-terminal TMD segments are critical for the presentation of the cleavage site region to the active site and the formation of a cleavage-competent state.

## Results

To investigate the mechanism governing substrate selection we compared a series of natural and designed γ-secretase substrates in order to relate the site-specific conformational flexibility of their TMD helices to the efficiency and specificity of cleavage in detergent and a cellular membrane (Fig. 1).

**Biphasic DHX kinetics diagnose highly flexible regions within the TM helices of γ-secretase substrates.** In order to examine the role of local helix flexibility for γ-secretase cleavage, we determined the stability of amide H-bonds by DHX in different TMD model constructs and correlated them to their cleavage efficiency and sequence-specificity in vitro and in cellulo. For DHX analysis, we used synthetic peptides where the hydrophobic TMD residues are flanked by Lys triplets (Supplementary Table 1). Similar to our previous analysis of various substrate TMD peptides[8–11,27–29], DHX kinetics of exhaustively (>95%) deuterated peptides were measured at 20 °C and pH 5 in 80% trifluoroethanol (TFE). The polarity of TFE roughly matches that within the solvated interior of a protein[30] and is therefore thought to mimic the aqueous environment within presenilin[31]. Gas-phase fragmentation after different periods of DHX, yielded residue-specific DHX kinetics. From the DHX kinetics we derived the corresponding amide exchange rate constants $k_{exp}$ (Supplementary Table 2) leading to the distributions of the respective free

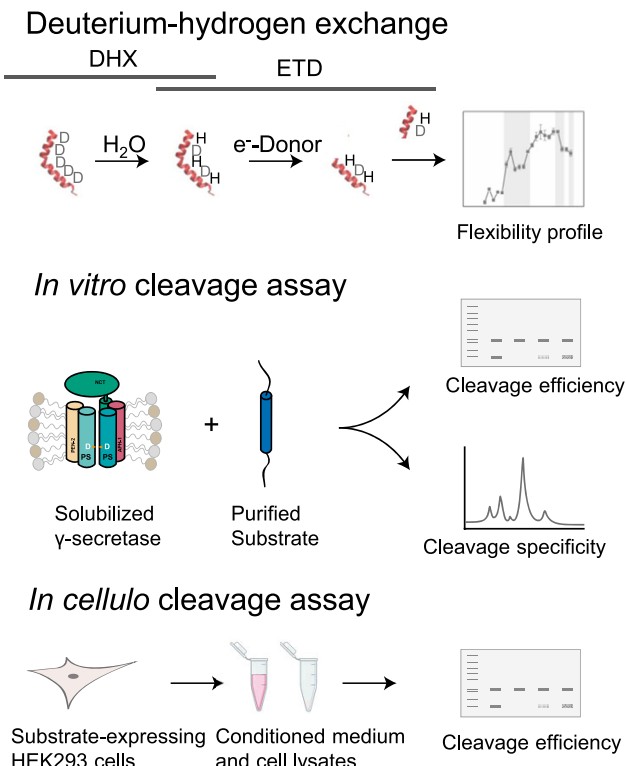

### Deuterium-hydrogen exchange

Flexibility profile

### In vitro cleavage assay

Solubilized γ-secretase + Purified Substrate → Cleavage efficiency / Cleavage specificity

### In cellulo cleavage assay

Substrate-expressing HEK293 cells → Conditioned medium and cell lysates → Cleavage efficiency

**Fig. 1 Outline of experimental strategy.** The mechanism of cleavage was delineated based upon deuterium-hydrogen exchange analysis, combined with in vitro and in cellulo cleavage assays.

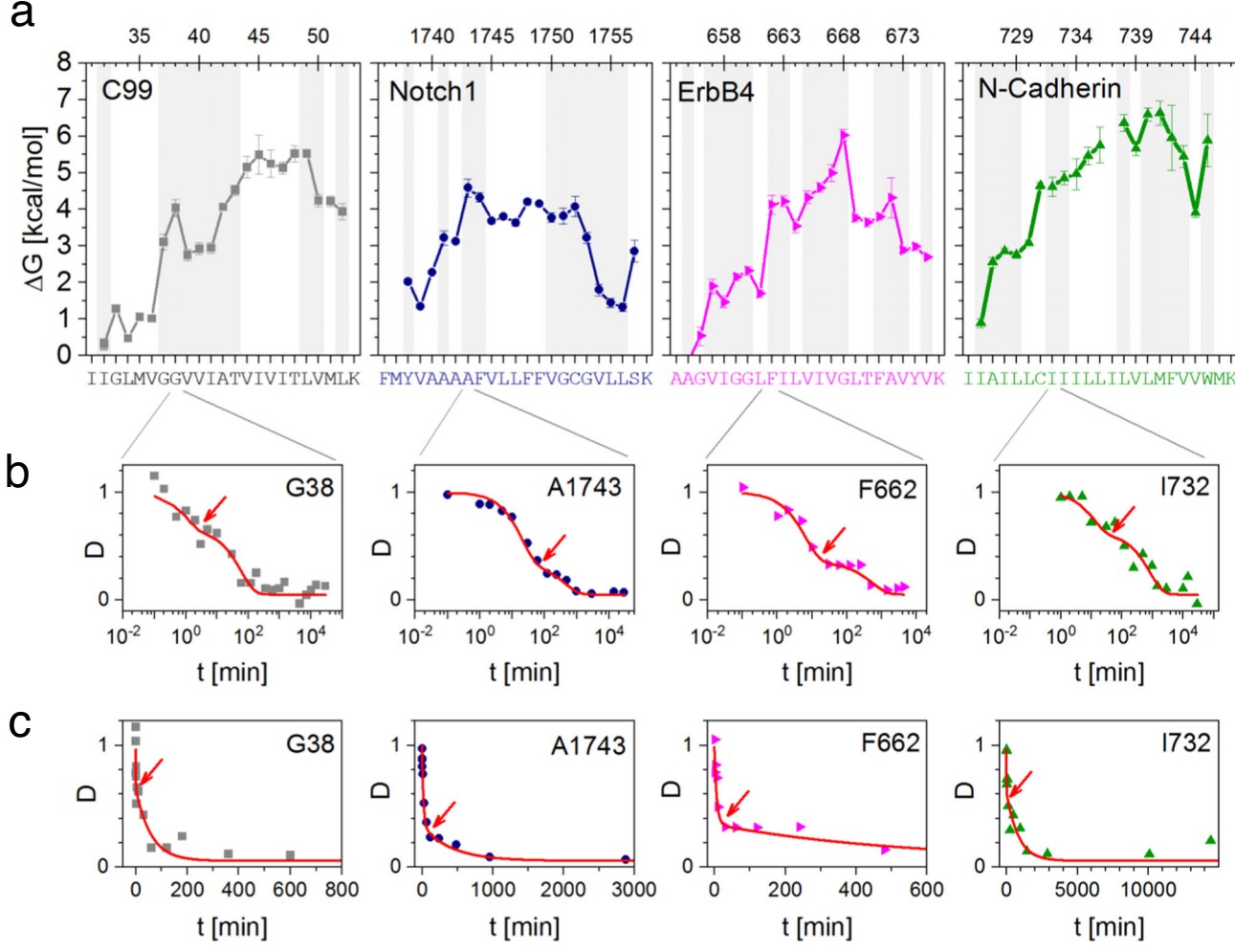

**Fig. 2 Comparing helix flexibilities of various γ-secretase substrate TMDs. a** Comparison of amide H-bond stabilities ΔG calculated from $k_{exp}$ values given in Supplementary Fig 1. Areas of biphasic DHX are shaded. Sequence positions are given above the graphics (Aβ numbering is used for C99) and the complete TMD peptide sequences are given in Supplementary Table 1. No values can be shown for N-Cadherin I737 due to poor fragmentation efficiency and/or overlap of isobaric fragments. Error bars correspond to standard confidence intervals (calculated from the errors of fit in $k_{exp}$ determination, in some cases smaller than the symbols, $N = 3$ independent DHX reactions). **b**, **c** Exemplary DHX kinetics that is fitted with a biexponential decay function at a logarithmic **b** or linear **c** scale. Note that linear time axes are truncated such as to better visualize the intersections between fast and slow regimes of exchange (arrows). Data are reproduced from Supplementary Fig 1.

energy changes ΔG that are associated with the disruption of amide H-bonds[27]. These distributions are designated 'flexibility profiles' (Figs. 1 and 2a).

Previously, the flexibility profile of the C99 TMD was obtained by fitting residue-specific DHX kinetics with a monoexponential decay function[9,10]. Here, a critical re-examination of the C99 kinetics, which was supplemented with additional data, revealed that the exchange kinetics from G37 to A42 and near both helix termini are fitted more appropriately with a biexponential function (Fig. 1 and S1). For example, Fig. 2 shows biphasic exchange at G38 at logarithmic (part b) or linear (part c) time scales. We propose that the fast phase reflects a superposition of uncorrelated exchange (EX2 regime) with rare correlated exchange reactions (EX1 regime) in a mixed EX1/EX2 mode[32,33]. By contrast, the slow phase results from uncorrelated exchange only, as detailed in Supplementary Note 1 and Supplementary Fig 2. In this model, correlated DHX signifies amide H-bonds being open long enough for two or more exchange events to occur simultaneously, such as at frayed helix termini and bent helices. By contrast, EX2-type uncorrelated exchange occurs at amides where the formation of H-bonds after local unfolding is

much more rapid than DHX; this permits the determination of ΔG values that describe the free energy change of H-bond opening at a given amide[34]. In cases of biphasic DHX, therefore, we determined ΔG values from the slow phase.

Figure 2a shows that amides from G37 to A42, containing the helix bend, are described by lower ΔG values (≈3–4 kcal/mol) than most other amides (≈4–5.5 kcal/mol). At the same time, G37-A42, L49-V50, and both helix termini exhibit biphasic DHX (Fig. 2a, shaded regions and Supplementary Fig 1). As noted above, this suggests the contribution of a mixed EX1/EX2 regime as another hallmark of a locally unstable helix. In order to relate the exchange behavior to helix geometry, we compared the DHX data to the lengths and angles of H-bonds within the TM helix in the C99 NMR structure[6]. Supplementary Fig 3 shows that none of the amides from G38 through I41 exhibits strong H-bonding. G37 and A42 bordering this segment appear to form strong α-helical H-bonds in the structure. The biphasic exchange at these positions may reflect correlated DHX together with amides of G38 or I41, respectively.

In order to examine the potential occurrence of highly flexible regions within the TM helices from γ-secretase substrates other

than C99, we also investigated well-established substrate TMDs with different primary structures and biological roles[35] by DHX. The flexibility profiles of the TMD helices of ErbB4 and N-Cadherin were determined and compared to those of C99 and Notch1[28]. In all cases, ΔG values tended to be lower within TM-Ns than within TM-Cs. In addition, all TMDs contain internal regions and helix termini exhibiting biphasic exchange kinetics (Fig. 2a).

Taken together, biphasic DHX kinetics appear to diagnose unstable regions with weak amide H-bonding at massively deformed sites within the C99 TM helix, such as at the bend at the $V_{36}G_{37}G_{38}V_{39}$ motif, as well as at frayed helix termini. By analogy, the analysis of other γ-secretase substrate TMDs also suggests highly flexible regions including their TM-N helices that may thus be a widespread feature of γ-secretase substrates.

**Introducing a hinge into the N-terminal half of a non-substrate poly-Leu TMD facilitates its cleavage.** Assuming that a substrate TMD helix must be conformationally flexible[12,13], we reasoned that a rigid poly-Leu sequence[36], denoted in the following as pL, may resist cleavage. The helix-stabilizing effect of Leu[37,38] at position (i) is ascribed to favorable interactions of its large and flexible side chain with side chains of its (i ± 3, 4) neighbors along an α-helix[38]. Our attempts to assign site-specific exchange rates to a pL helix tagged with Lys triplets failed due to massively overlapping fragment patterns generated by ETD of this symmetric sequence. However, the global rigidity of an oligo-Leu helix was previously demonstrated by very slow overall DHX[36] and H-bond stability within the oligo-Leu regions of peptides pL-GG, pL-VGGV, or pL-A9 described below (Fig. 3a) approaches ~6 kcal/mol, thus attesting to high local stability.

To assess the potential cleavability of a rigid pL helix by γ-secretase, we used recombinant C99 constructs based on the C100-His$_6$ protein[39]. We first compared wt C99 to the pL construct, a corresponding chimera holding a 24-residue poly-Leu sequence in place of the natural TMD using a well-established in vitro assay[40]. The generated AICD and Aβ peptides (we also refer to Aβ peptides for peptides generated by the pL variants) were analyzed via immunoblotting (Fig. 3b–e) and by MALDI-TOF mass spectrometry (MS) (Fig. 3f, g)[41]. Generation of both AICD and Aβ from pL was strongly reduced to levels of ~6% or ~3% of C99, respectively (Fig. 3b–e), suggesting that pL is effectively a non-substrate of γ-secretase.

C99 cleavage is a sequential process starting at alternative ε-sites and then proceeding to ζ-sites and γ-sites. Depending on the chosen ε48 or ε49 site, cleavage produces a 51- or 50-residue-long APP intracellular domain (AICD) plus various Aβ peptides, typically ranging from Aβ37-Aβ42, as end products of the stepwise cleavages, including Aβ40 as predominant form[1,42]. For wt C99, the MS pattern of AICD products showed the two characteristic major cleavage products with the predominance of the ε49-cleaved (AICD50) over the ε48-cleaved (AICD51) form, thus confirming the known preferential cleavage at ε49 (Fig. 3f). In line with this, the intensity of the MS signal for Aβ40 exceeded that of Aβ42 and of a number of smaller cleavage products (Fig. 3g). Importantly, neither AICD nor Aβ was detected in the case of pL, thus establishing the poly-Leu TMD as a non-substrate TMD.

Using pL as a template for examining the importance of C99-derived sequence motifs, we next asked to which extent the reintroduction of the $G_{37}G_{38}$ or $V_{36}G_{37}G_{38}V_{39}$ hinge motifs would confer both conformational flexibility and cleavability to pL. DHX analysis of the hybrid pL-GG and pL-VGGV peptides revealed (i) reduced H-bond stabilities near the glycines relative to pL-A9 (pL-GG, ΔG reduced by ~1-1.5 kcal/mol; pL-VGGV,

~2 kcal/mol), and (ii) biphasic DHX within the LGGL and VGGV motifs, i.e., shifted towards the N-terminus compared to C99 (Fig. 2a and S4). The poly-Leu/poly-Ala hybrid pL-A9 was used to assess the stability of the N-terminal half of a polyLeu helix. The engineered hinge regions thus appear to be somewhat less pronounced than within the C99 TMD, and more flexible in pL-VGGV than in pL-GG.

Interestingly, cleavage of the corresponding C99-based pL-GG and pL-VGGV constructs produced AICD and Aβ fragments at levels of 33% and 40% (AICD) and 19% and 26% (Aβ) of wt C99, respectively (Fig. 3b–e). Thus, introducing a hinge into the poly-Leu TMD partially restores cleavability. For both constructs, pL-GG, and pL-VGGV, initial cleavage remained highly site-specific with ε48-cleaved and ε49-cleaved AICDs predominating, albeit the site of preferential cleavage was shifted from ε49 to ε48 (Fig. 3f). Surprisingly, abundant Aβ species, such as Aβ40, were not detected with these constructs. For the somewhat better cleavable pL-VGGV, the only species found were peptides ≤Aβ38, with Aβ34 as the major form (Fig. 3g). Compared to wt C99, this indicates much more efficient processing of pL-GG and pL-VGGV proteins across the canonical γ-sites.

Figure 3 h summarizes the data by connecting the strengths of single H-bond opening, as visualized by heat maps, to regions showing biphasic DHX and the location and usage of the cleavage sites. It illustrates how introducing a hinge region into TM-N enhances cleavability, albeit with altered cleavage site usage, of an otherwise rigid and uncleavable poly-Leu TMD.

**Elevating flexibility within the C-terminal half of a TMD does not promote cleavage.** Having shown that the flexibility-conferring diglycine hinge motifs in a poly-Leu TM-N can partially restore cleavage, we next probed the potential impact of enhancing flexibility in the C-terminal half. To this end, we replaced two Leu residues in pL-VGGV by Gly at positions 48 and 49 that are equivalent to the wt C99 ε-cleavage sites. On the one hand, the resulting pL-VGGV/εGG construct exhibits (i) H-bond stabilities near the glycines that are reduced by ~1–3 kcal/mol relative to pL-VGGV, and (ii) biphasic DHX from positions 46 through 50 (Fig. 4a). On the other hand, in vitro cleavage of the corresponding C99 chimera did not indicate enhanced AICD or Aβ production relative to the parental pL-VGGV (Fig. 4b–e) indicating that artificially enhancing helix flexibility around the initial cleavage sites does not promote cleavage. Further, the AICD mass spectrum obtained from pL-VGGV/εGG was similar to that of pL-VGGV with a major peak resulting from cleavage at ε48 (Fig. 4f). The analysis of the Aβ species revealed minor amounts of Aβ35, Aβ37, and Aβ38 peptides in addition to Aβ34, again being reminiscent of pL-VGGV (Fig. 4g). Again, Fig. 4h summarizes the relationships between DHX profiles and cleavage specificity.

That enhancing helix flexibility around the initial cleavage sites does not promote cleavage of C99 by γ-secretase seems to contradict earlier observations where the double mutation I47G/T48G elevated initial C99 cleavage, whereas I47L/T48L strongly inhibited cleavage[24]. The kinetic analysis of the cleavage of these substrates suggested that those changes resulted from altered reaction velocities ($V_{max}$) of the bound substrates, rather than from changes in binding affinity ($K_m$). As those results suggested an apparent link between helix flexibility of TM-C and cleavability, we studied the conformational flexibility of these mutant helices by DHX. As expected, I47G/T48G reduces amide H-bond stability around the mutated positions by up to ~2.5 kcal/mol (Supplementary Fig 6). Contrary to our expectation, however, I47L/T48L also slightly destabilizes amide H-bonds by up to ~1 kcal/mol, for potential reasons discussed below. These

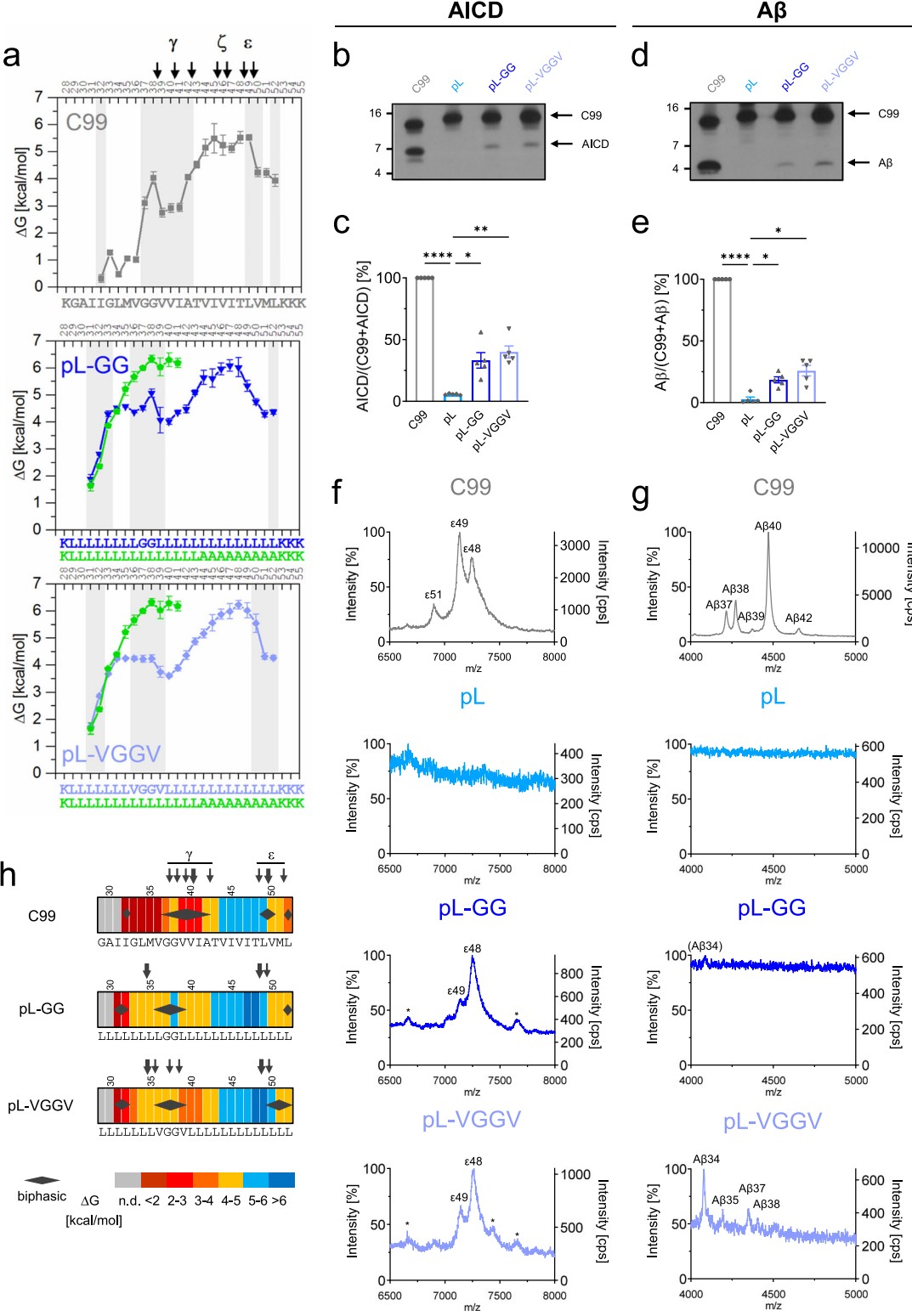

findings again challenge the idea that helix flexibility around the ε-cleavage sites scales with C99 cleavability.

**Combining the N-terminal hinge with the natural cleavage region restores cleavability.** In another attempt to restore C99-level cleavability, we reintroduced V44 to L52, a stretch of sequence encompassing the ε-cleavage sites (termed cr). DHX showed that the helix flexibility of pL-VGGV/cr roughly matched that of pL-VGGV (Fig. 5a). Interestingly, cleavage assays of

pL-VGGV/cr clearly showed an increase in AICD and Aβ production relative to pL-VGGV, up to a level of 82% or 68%, respectively, of C99 (Fig. 5b–e). Including the cleavage region, however, did not restore the C99 pattern of AICD fragments. Rather, the usage of ε-sites was similar to pL-VGGV, i.e., mainly at ε48 (Fig. 5f). The Aβ pattern of pL-VGGV/cr also did not match that of wt C99 (Fig. 5g). Again, a mixture of shorter Aβ peptides indicated an increased processivity, reminiscent of pL-VGGV and pL-VGGV/εGG. The cleavage efficiency of pL-cr, a

**Fig. 3 Improving helix flexibility within the N-terminal half of a non-substrate poly-Leu TMD partially restores cleavage. a** Amide H-bond stabilities ΔG calculated from $k_{exp}$ values given in Supplementary Fig 5. Regions of biphasic DHX are shaded and canonical cleavage sites are indicated. C99 data are reproduced from Fig. 1 for comparison. Only the poly-Leu part of pL-A9 is shown where data quality sufficed for ΔG calculation. Error bars correspond to standard confidence intervals (calculated from the errors of fit in $k_{exp}$ determination, in some cases smaller than the symbols, $N = 3$ independent DHX reactions). **b, d** Cleavage efficiency of the different constructs after incubation with CHAPSO-solubilized HEK293 membrane fractions at 37 °C. Levels of AICD (**b**) and Aβ (**d**) were subsequently analyzed by immunoblotting. Quantification of AICD **c** and Aβ **e** levels, values shown as percent of wt C99 (=100%). The TMD sequences of C99-based constructs are given in **a**. Data are represented as means ± SEM, $N = 5$. Statistical significance was assessed using one-way ANOVA with Dunnett's multiple comparison test and pL as a control condition (*$p < 0.05$, **$p < 0.01$, ***$p < 0.001$, ****$p < 0.0001$). **f** Representative MALDI-TOF spectra from three independent measurements show the different AICD fragments generated for the various constructs. The intensities of the highest AICD peaks were set to 100%. Additionally, the counts per second (cps) is shown on the right y-axis. * denotes unspecific peaks. Calculated and observed masses for each peak can be found in Supplementary Table 3. **g** Representative MALDI-TOF spectra from two independent measurements show the different Aβ fragments generated for the various constructs. The intensities of the highest Aβ peaks were set to 100%. Calculated and observed masses for each peak can be found in Table S4. **h** Heat maps summarizing the color-coded ΔG values of single H-bond openings (n.d. = not determined), the occurrence of biphasic amide DHX (diamonds), and experimentally determined γ-secretase cleavage sites (arrows).

derivative lacking the VGGV motif, was similar to that of pL-VGGV (Fig. 5b–e). Thus, the cleavage region exhibits some cleavability even without the VGGV hinge. While the patterns of pL-cr AICD fragment matched those of pL-VGGV, it produced mainly Aβ36, indicating processivity intermediate between C99 and pL-VGGV/cr (Fig. 5f, g). Unfortunately, we were unable to interpret DHX data of the pL-cr peptide as we could not identify a sufficient number of fragments after gas-phase fragmentation.

Taken together, grafting residues V44 to L52 including both ε-sites onto the oligo-Leu TMD markedly enhanced cleavage. Only in cooperation with the VGGV hinge, however, does this region confer cleavability that is now close to that of wt C99. The pronounced rank order of cleavability (C99 > pL-VGGV/cr > pL-VGGV) contrasts the similarity of these sequences in terms of helix flexibility around the ε-cleavage sites (Fig. 5h).

**C99 cleavage in cellulo supports conclusions drawn from in vitro analysis.** In order to investigate the role of the different sequence motifs in a cellular membrane with its complex mixture of natural lipids and proteins, we also performed cleavage assays in HEK293 cells transfected with the corresponding C99 derivatives. The extent of γ-cleavage by the cell's endogenous γ-secretase was assessed by determining the levels of secreted Aβ peptides via immunoblotting. As shown in Fig. 6, replacing the C99 TMD by the poly-Leu sequence virtually abolished Aβ secretion in the cell-based assay, corroborating the poly-Leu TMD as a non-substrate. A control flow cytometric experiment revealed that, compared to wt C99, the pL construct is slightly less, but still efficiently expressed at the cell surface (Supplementary Fig 7), thus ruling out intracellular retention of pL as a potential source of its near-complete resistance to cleavage. Inserting the G37G38 or V36G37G38V39 motifs again partially restored cleavage to levels of 13% to 22%, respectively, thus confirming the importance of the TM-N hinge region. A similar increase in Aβ production was also seen upon grafting the cleavage region (V44 to L52) onto poly-Leu. Importantly, a further elevation of Aβ secretion, up to a level similar to wt C99, was achieved by combining the hinge and the cleavage region in the pL-VGGV/cr construct. Similar to the situation in the cell-free assays, a GlyGly pair in the TM-C (pL-VGGV/εGG) did not promote Aβ production above that of pL-VGGV.

Together, the in cellulo cleavage data confirm that a poly-Leu TMD is a non-substrate for γ-secretase. Importantly, the impacts of the hinge and the cleavage region on processing by γ-secretase are similar in cell-based and cell-free assays.

## Discussion
The main objective of this study was to illuminate the role and cooperative behavior of different parts of a TMD in defining its

cleavability by γ-secretase. Our overarching aim was to better understand how γ-secretase discriminates substrates from non-substrates, this is also a major unresolved issue with other intramembrane proteases. To this end, we examined hybrid sequences between C99 and an artificial non-substrate, denoted pL, holding a rigid and featureless poly-Leu TMD. pL is virtually uncleaved in vitro and in cellulo, despite its permissively short extracellular domain and the presence of the C99 juxtamembrane domains. This non-substrate served as a template to identify critical features of a substrate TMD.

First, grafting the C99-derived G37G38 or V36G37G38V39 motifs onto pL introduced a highly flexible site within TM-N, as indicated by low H-bond strengths and local biphasic DHX kinetics reminiscent of the di-glycine hinge within the C99 TMD. While the fast phase of biphasic amide exchange is proposed to result from a superposition of rare correlated exchange events with the dominating non-correlated exchange at early stages of the reaction, thus artificially accelerating the exchange rate, the slow phase is thought to correspond to the kinetics of non-correlated exchange. As such, the occurence of biphasic DHX complements the observation of low H-bond strength in diagnosing massive helix deformations. Biphasic DHX had previously been associated with fluctuating conformers of soluble proteins[32,33]. To our knowledge, this is the first report of biphasic DHX within TM helices where correlated and non-correlated exchange may take place in parallel at highly unstable sites.

While grafting G37G38 onto pL partially restores cleavability, the V36G37G38V39 motif confers not only stronger flexibility than G37G38 but also more effective cleavage. This gain-of-function complements previous mutational studies where mutating G38 had impaired cleavage[10] and provides further proof that a conformationally flexible TM-N is important for C99 cleavage by γ-secretase. MD simulations of the enzyme/C99 complex had suggested previously that TM-N bending enables the C99 TM helix to enter presenilin and access its catalytic cleft[7,43]. By analogy to C99, helix bending may promote the translocation of other substrate TMDs from sites of initial contact towards the catalytic center of γ-secretase. This is supported by the detection of weak H-bonding and biphasic DHX within the TMDs of well-known γ-secretase substrates other than C99. In the case of Notch1, the region exhibiting correlated H-bond openings overlaps partially with the tetra-Ala motif that is distorted in the γ-secretase/Notch1 structure[17]. Although NMR spectroscopy in the helix-stabilizing hydrophobic environment of a detergent micelle had previously indicated a rather straight Notch1 TM helix, MD simulations had suggested helix unwinding at L1747, which is part of the flexible region detected here[44]. With ErbB4 and N-Cadherin, the identification of correlated H-bond openings coincides with the abundance of helix-destabilizing Gly and/or β-

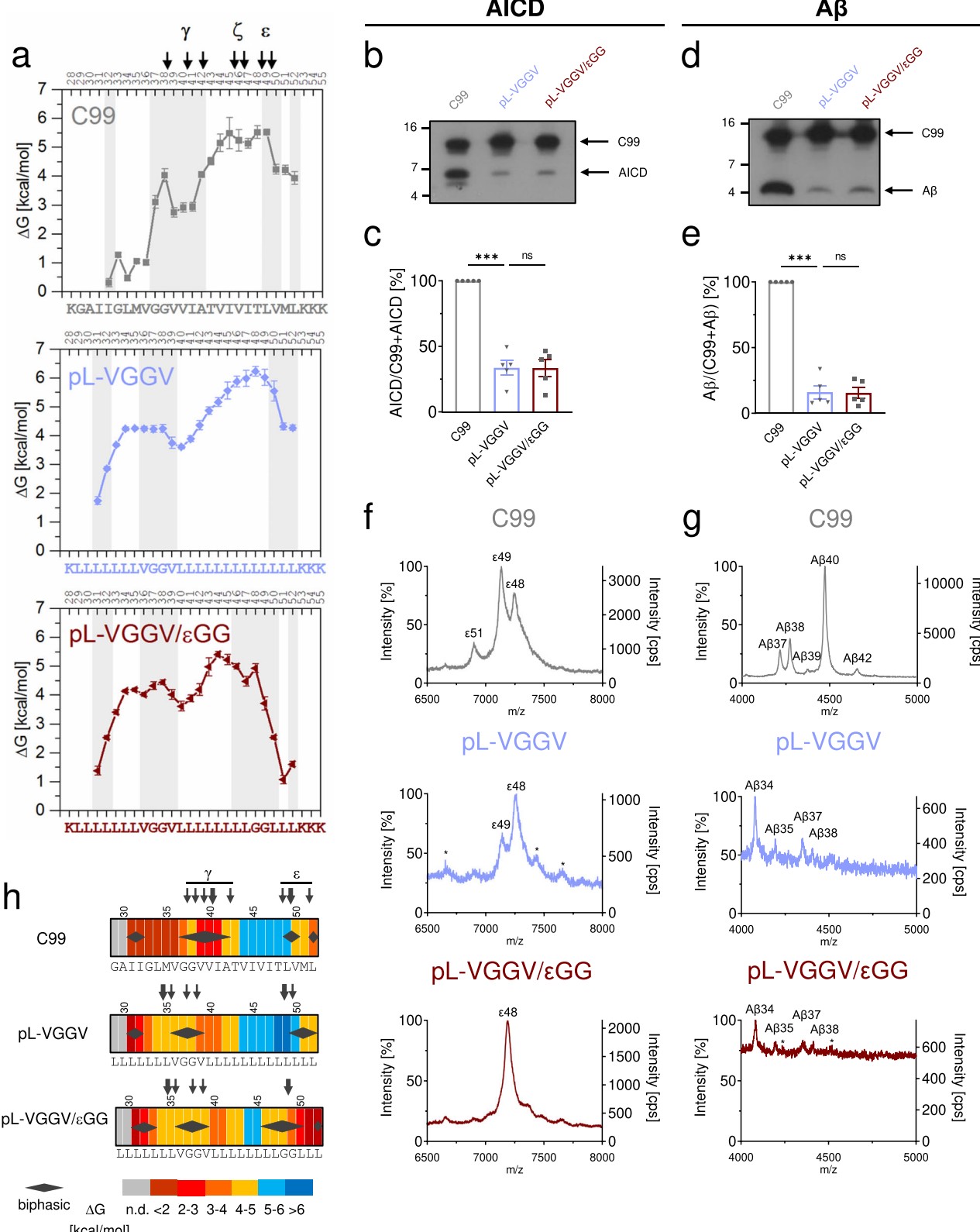

branched residues. Altogether, these findings let us propose that substantial conformational flexibility within TM-N, possibly associated with dynamic helix bending, may be one crucial requirement for γ-secretase substrates. Conformational flexibility may also facilitate the processing of substrates of other intra-membrane proteases including site-2-protease, signal peptide peptidase (SPP) and signal peptide peptidase-like (SPPL)

proteases, as well as rhomboid proteases (reviewed in ref. [12,13]). Recent examples include the SPP substrate Xbp1u where stabilizing the TM helix by Leu abolished cleavage[27]. NMR spectroscopy recently uncovered a slight bend at the $A_{42}G_{43}A_{44}$ motif in the center of the TM helix of tumor necrosis factor α (TNFα), an SPPL2a substrate. Replacing $A_{42}G_{43}A_{44}$ by Leu stabilized the helix and strongly decreased cleavage at a downstream site of the

**Fig. 4 The flexibility of the cleavage region does not determine cleavage efficiency. a** Amide H-bond stabilities ΔG calculated from $k_{exp}$ values given in Supplementary Fig 4. C99 and pL-VGGV data are reproduced from Fig. 3 for comparison. Regions of biphasic DHX are shaded and canonical cleavage sites are indicated. Error bars correspond to standard confidence intervals (calculated from the errors of fit in $k_{exp}$ determination, in some cases smaller than the symbols, $N = 3$ independent DHX reactions). **b, d** Cleavage efficiency of the different constructs after incubation with CHAPSO-solubilized HEK293 membrane fractions at 37 °C. Levels of AICD **b** and Aβ **d** were subsequently analyzed by immunoblotting. Quantification of AICD **c** and Aβ **e** levels, values shown as percent of wt C99 (=100%). Data are represented as means ± SEM, $N = 5$. Statistical significance was assessed using one-way ANOVA with Dunnett's multiple comparison test and pL-VGGV as a control condition (*$p < 0.05$, **$p < 0.01$, ***$p < 0.001$, ****$p < 0.0001$). **f, g** Data for C99 and pL-VGGV are reproduced from Fig. 3 for comparison. **f** Representative MALDI-TOF spectra from three independent measurements show the different AICD fragments generated for the various constructs. The intensities of the highest AICD peaks were set to 100%. Additionally, the counts per second (cps) is shown on the right y axis. * denotes unspecific peaks. **g** Representative MALDI-TOF spectra from two independent measurements show the different Aβ fragments generated for the various constructs. The intensities of the highest Aβ peaks were set to 100%. Additionally, the counts per second (cps) is shown on the right y axis. * denotes unspecific peaks. **h** Heat maps summarizing the color-coded ΔG values of single H-bond openings (n.d. = not determined), the occurrence of biphasic amide DHX (diamonds), and experimentally determined γ-secretase cleavage sites (arrows).

TMD[45]. Likewisely, increased or decreased TMD helix flexibility facilitated or impeded SPPL3-dependent shedding, respectively, of its substrate GnTV[29].

A second crucial finding reported here relates to a hypothetical role of conformational flexibility within TM-C. As noted previously[9,10], the C99 TM-C helix is rather rigid in terms of amide H-bond stability although the detection of biphasic DHX at L49 and V50 suggests slow helix refolding at this site. Slow refolding within the C99 TM-C would not be surprising, as it contains an overabundance of β-branched residues that are known to destabilize helices[37,38]. We found that TM-C is important as residues V44-L52 confer a level of cleavability to pL-cr comparable to that conferred by the $V_{36}G_{37}G_{38}V_{39}$ motif in pL-VGGV. Does the importance of TM-C for cleavage efficiency rest on its conformational flexibility? We consider this an unlikely scenario since a di-glycine pair within the TM-C of pL-VGGV/εGG strongly enhances flexibility but not cleavability. Moreover, we cannot confirm the previously presumed opposite impact of two C99 double mutations (I47G/T48G, I47L/T48L) on conformational flexibility around the ε-sites, thus challenging a proposed dependence of cleavability on local helix flexibility[24]. Although the Gly residues indeed destabilize the I47G/T48G TMD near the mutated sites, as expected, the Leu residues of I47L/T48L also have a slight local destabilizing effect. C99 TM-C destabilization by Leu may result from removing the side-chain/main-chain H-bond that extends from T48 to V44 in the wt C99 TM-C and stabilizes the helix[46]. Possibly, the strongly reduced cleavability of the I47L/T48L mutant is related to steric hindrance of cleavage by the large Leu side chains. That TM-C helix flexibility does not determine the efficiency of initial cleavage is also suggested by previous findings on other substrates/enzyme systems. In the case of Notch1, the initial S3 cleavage site is formed by G1743 and V1744. Exchanging V1744 by either Leu or Gly strongly compromised cleavage[26]. Also, helix-destabilizing C49P or H52P mutations directly at the cleavage site of the TNFα TMD did not enhance its cleavage by SPPL2a[45]. The finding that increasing the level of helix flexibility within TM-C does not scale with cleavability is initially surprising given that the region from I47 to V50 of the C99 TMD is unfolded within γ-secretase and residues M51-K54 form a hybrid β-sheet with presenilin[18]. This suggests that TM-C unfolding around the scissile site is indeed required for cleavage. Since, however, TM-C flexibility does not correlate with the cleavability of the substrate TMDs investigated here, helix unfolding around cleavage sites may not constitute a rate-limiting step in intramembrane proteolysis. Indeed, helices in aqueous solution unfold at nanosecond timescales[47], which is orders of magnitude faster than notoriously slow intramembrane proteolysis[48–50] and thus unlikely to limit its rate.

Our third crucial finding is that initial cleavage within the oligo-Leu sequence of pL-GG and pL-VGGV takes place at the

canonical positions 48 and 49 (although cleavage preference is shifted to residue 49 while C99 is mainly cut at ε49). It appears therefore as if the selection of both ε-sites is largely independent of the natural C99 TMD sequence. Rather, the presentation of the scissile sites to the catalytic residues may mainly be defined by the overall geometry of the substrate-TMD/enzyme complex. Further, the formation of smaller Aβ species at the expense of major ones, such as Aβ40, from our model substrates may indicate that the sites of carboxyterminal trimming are shifted towards the N-terminus. Once the initial cleavage has taken place, the conformational rigidity of the oligo-Leu sequence in between cleavage region and $V_{36}G_{37}G_{38}V_{39}$ hinge may induce skipping of the γ- and ζ-cleavage sites that are characteristic of C99, thus producing the shorter Aβ fragments observed here. In an alternative model, the processivity of carboxyterminal trimming of our hybrid constructs may be greatly enhanced such that Aβ40 and Aβ42 are efficiently turned over to a range of shorter Aβ species. Similarly, C99 mutants containing stretches of Phe residues were preferentially cleaved at G38 in the previous work[51].

What are the implications of our results for the mechanism of substrate selection? Intramembrane proteolysis takes minutes to hours, as indicated by low catalytic constants[48,50,52]. Catalytic constants comprise the kinetics of all reaction steps downstream of initial substrate binding[53]. We propose that at least two reaction steps limit cleavage after binding. One rate-limiting process may correspond to translocation of a substrate from an exosite at the interface of the lipid bilayer and presenilin towards its catalytic cleft were the water molecules required for proteolysis are sequestered within the interior of the protein. Translocation beyond sterically obstructing TMDs of presenilin and the loops connecting them is expected to be facilitated by the conformational flexibility of a substrate TM-N. Another rate-limiting process might correspond to the formation of a cleavage-competent state once the TMD has reached the catalytic cleft. Although the flexibility of the TM helix around the ε-site does not determine cleavability, as indicated by the present data, one may envision that the C99 V44 – L52 region is crucial for the combined process of docking the unfolded sequence near the catalytic residues and/or the formation of the tripartite β-sheet with residues near the presenilin TMD7 N-terminus and within the TMD6/TMD7 connecting loop[54]. Although the formation of β-sheet from covalently connected strands may take only tens of microseconds[55], the assembly of sheet from disconnected strands may take minutes, as exemplified by aggregation-driven sheet formation in lysozyme[56]. In sum, at least two slow processes that involve substrate TM-N and TM-C, respectively, appear to limit the kinetics of intramembrane proteolysis. The presence of a conducive TM-N and TM-C that functionally cooperate may thus distinguish substrate TMDs from non-substrate TMDs (Fig. 7).

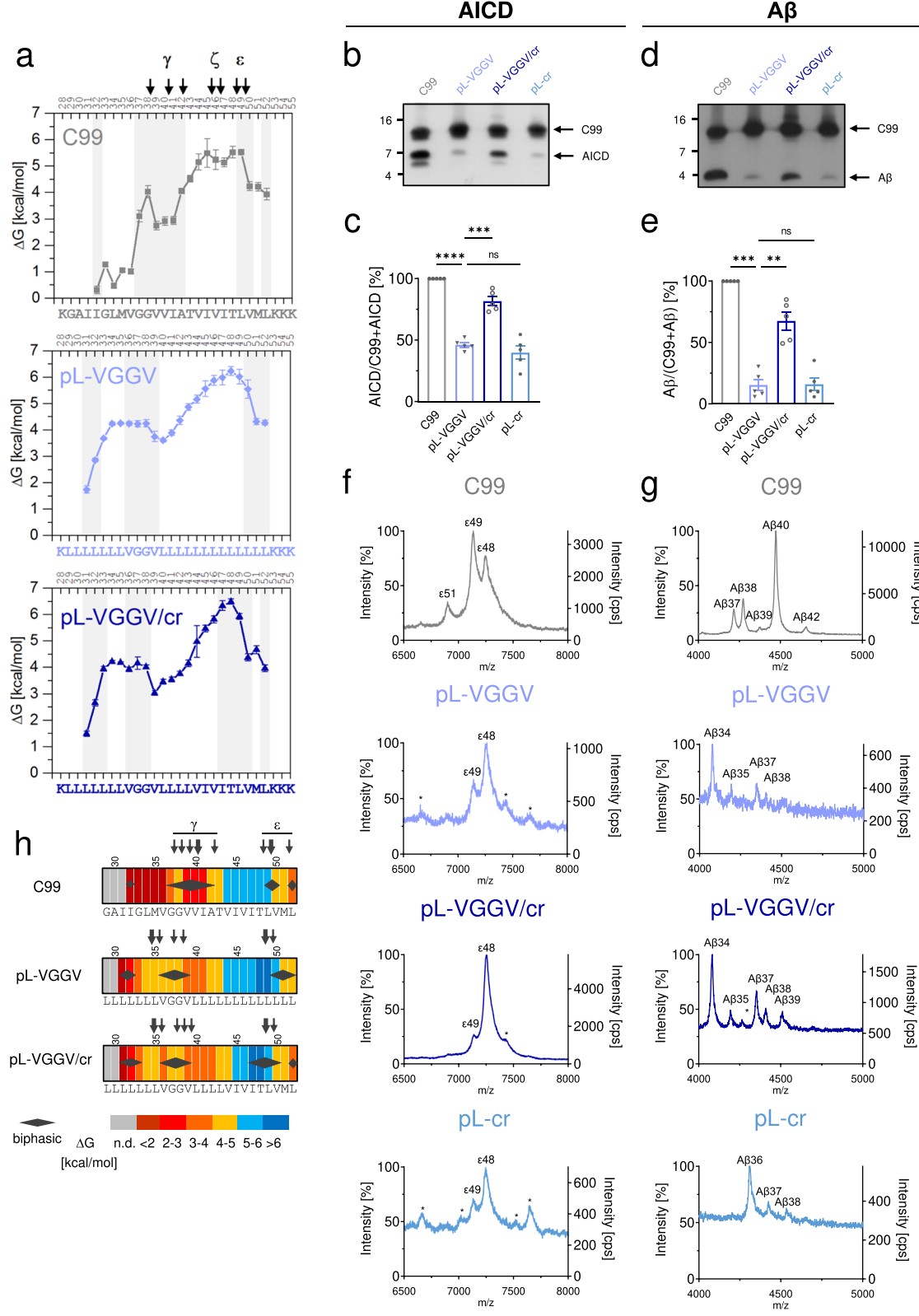

## Methods

**Peptide synthesis**. Peptides were synthesized by Fmoc chemistry by PSL, Heidelberg, Germany, and purified to >90% purity as judged by mass spectrometry. All other chemicals were purchased from Sigma-Aldrich Co. (St. Louis, Missouri, USA).

**Deuterium-hydrogen exchange by MS/MS**. All mass spectrometric experiments were performed on a Synapt G2 HDMS (Waters Co., Milford, MA) and

measurements were taken from distinct samples. Samples were injected from a 100 μL Hamilton gas-tight syringe via a Harvard Apparatus 11 Plus with a flow rate of 5 μL/min. Spectra were acquired in a positive-ion mode with one scan per second and a 0.1 s interscan time. Solutions of deuterated peptide (100 μM in 80% (v/v) d1-trifluoroethanol (d1-TFE) in 2 mM $ND_4$-acetate) were diluted 1:20 with protonated solvent (80% (v/v) TFE in 2 mM $NH_4$-acetate, pH 5.0) to a final peptide concentration of 5 μM and incubated at 20.0 °C. Exchange reactions were quenched by cooling on ice and lowering the pH to 2.5 by adding 0.5% (v/v)

**Fig. 5 The native sequence around ε-sites cooperates with the hinge to restore cleavability. a** Amide H-bond stabilities ΔG of pL-VGGV/cr calculated from $k_{exp}$ values given in Supplementary Fig 4. C99 and pL-VGGV data reproduced from Fig. 3 for comparison. Regions of biphasic DHX are shaded and canonical cleavage sites are indicated. Error bars correspond to standard confidence intervals (calculated from the errors of fit in $k_{exp}$ determination, in some cases smaller than the symbols, $N = 3$ independent DHX reactions). **b**, **d** Cleavage efficiency of the different constructs after incubation with CHAPSO-solubilized HEK293 membrane fractions at 37 °C. Levels of AICD **b** and Aβ **d** were subsequently analyzed by immunoblotting. Quantification of AICD **c** and Aβ **e** levels, values shown as percent of wt C99 (100%). Data are represented as means ± SEM, $N = 5$. Statistical significance was assessed using one-way ANOVA with Dunnett's multiple comparison test and pL-VGGV as a control condition (*$p < 0.05$, **$p < 0.01$, ***$p < 0.001$, ****$p < 0.0001$). **f**, **g** Data for C99 reproduced from Fig. 3 for comparison. **f** Representative MALDI-TOF spectra from three independent measurements show the different AICD fragments generated for the various constructs. The intensities of the highest AICD peaks were set to 100%. Additionally, the counts per second (cps) are shown on the right y axis. **g** Representative MALDI-TOF spectra from two independent measurements show the different Aβ fragments generated for the various constructs. The intensities of the highest Aβ peaks were set to 100%. **h** Heat maps summarizing the color-coded ΔG values of single H-bond openings (n.d. = not determined), the occurrence of biphasic amide DHX (diamonds), and experimentally determined γ-secretase cleavage sites (arrows).

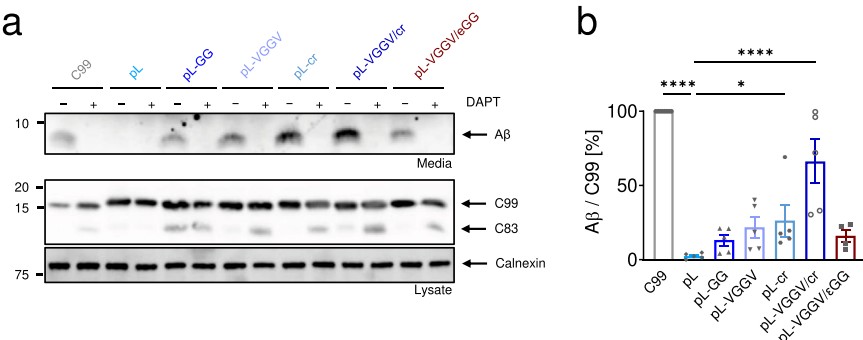

**Fig. 6 Influence of TMD sequence on Aβ production recapitulated in cellulo.** HEK293 cells were transiently transfected with the indicated wt and pL-based C99 constructs containing an N-terminal HA- and a C-terminal 2xFlag-tag. To control for specificity, transfected cells were treated with γ-secretase inhibitor DAPT (or vehicle DMSO) for 24 h. **a** HA-tagged Aβ secreted into the culture media, was detected via immunoblotting against HA. The cellular levels of the full-length tagged C99, as well as α-secretase cleavage product C83, were detected via immunoblotting against Flag. Calnexin was used as a loading control. **b** Quantification of HA-Aβ blots, like the exemplary one shown in **a**. Only vehicle conditions were quantified since no signals were detected with DAPT. The values represented were normalized to C99. Data are represented as means ± SEM, $N = 4$ for pL-VGGV/εGG, and $N = 5$ for the others. One-way ANOVA with Dunnett's multiple comparison tests were performed by taking pL as the control condition (*$p < 0.05$, **$p < 0.01$, ***$p < 0.001$, ****$p < 0.0001$).

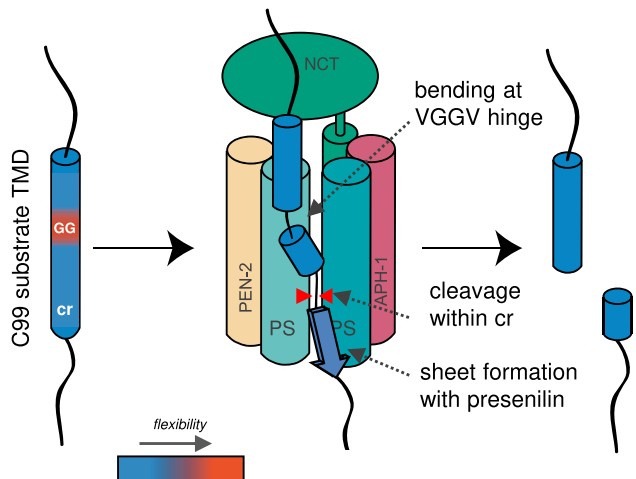

**Fig. 7** Graphical summary of the structural features enabling cleavage of a TMD by γ-secretase.

formic acid. Mass/charge ratios were recorded and evaluated as previously described[57]. For electron transfer dissociation (ETD), we used 1,4-dicyanobenzene as an electron donor and preselected 5+ charged peptides via MS/MS. Fragmentation of peptides was performed by accumulating ETD MS/MS scans over a 10 min scan time[11]. ETD-measurements were performed after different incubation periods (from 1 min to 3 d) where exchange had taken place at pH 5.0. Shorter (0.1 min, 0.5 min) and longer (5 d, 7 d) incubation periods were simulated by lowering the pH to 4.0 or elevating pH to 6.45, respectively, using matched time periods. The differences to pH 5.0 were taken into account when calculating the corresponding rate constants. We note that base-catalyzed exchange is responsible for at least 95% of the total deuteron exchange at ≥pH 4.0. The resulting ETD c' and z fragment spectra were evaluated using a semi-automated procedure (ETD FRAGMENT ANALYZER module of MassMap_2019-01_28_LDK Software[27]).

Monoexponential fitting of the data were done with Eq. (1) to calculate $k_{exp,DHX}$, which accounts for the concentration of the deuterated solution in the DHX-ETD assay of 5% (v/v)

$$D(t) = 0.95 \cdot e^{-k_{exp}t} + 0.05 \tag{1}$$

while biexponential fitting was done with Eq. (2)

$$D(t) = A \cdot e^{-k_{exp,A}t} + B \cdot e^{-k_{exp,B}t} + 0.05 \tag{2}$$

Where A and B are the population sizes of the deuterons with slower and faster exchange rates, $k_{exp,A}$ and $k_{exp,B}$, respectively, and $A + B = 0.95$. The decision between applying a mono- or biexponential fitting routine was based on Wilks' theorem using a $p$ value of 0.01 for monophasic behavior to assign biphasic ($p < 0.01$) or monophasic ($p > 0.01$) behaviour (see Supplemental Information for details).

The free energies ΔG required for H-bond opening were calculated from the smaller rate constant ($k_{exp,B}$) and $k_{ch}$ based on Eq. (3) based on Linderstrøm-Lang

theory[34], assuming EX2 conditions and a predominantly folded state[27].

$$\Delta G = -RT \ln\left(\frac{k_{exp,DHX}}{k_{ch} - k_{exp,DHX}}\right) \quad (3)$$

where $k_{ch}$ represents the sequence-specific chemical rate constants that were calculated using the program SPHERE (http://landing.foxchase.org/research/labs/roder/sphere/) (under the set conditions: D-to-H-exchange, reduced Cys, pH = 5.0, T = 20.0 °C).

**C99 substrate constructs**. All poly-L C99 constructs were generated by GenScript with a N-terminal signal sequence followed by a single N-terminal HA-tag and two C-terminal FLAG-tags in expression vector pcDNA3.1 to allow their direct use in cell-based assays. For cell-free assays, the constructs were initially recloned as PCR fragments into pQE60 (Qiagen) for C100-His6[39] to contain an N-terminal Met (M1 in C100) and a C-terminal His6 tag connected via a GSRS linker and then further recloned as NcoI/HindIII fragments into pET-21d(+) (Novagen) to increase expression yields.

**Production of recombinant proteins and γ-secretase in vitro cleavage assay**. Wt and pL-based C99 constructs were expressed in *E.coli* BL21(DE3)RIL or Rosetta(DE3) cells, respectively. All constructs were purified by Ni-NTA-agarose affinity chromatography. Briefly, induced cells were incubated for 3.5 h at 37 °C and sonicated. Following overnight urea-lysis [20 mM Tris/HCl (pH 8.5), 6 M urea, 1 mM CaCl₂, 100 mM NaCl, 1% Triton X-100, 1% SDS, with protease inhibitors], Ni²⁺-NTA-agarose was added and the samples were incubated for 2 h at room temperature with shaking. Following three washing steps, the substrate proteins were eluted with a buffer containing imidazole [50 mM Tris/HCl (pH 8.5), 300 mM NaCl, 0.2% SDS, 100 mM imidazole (pH 8.5)]. Purified constructs were incubated overnight at 37 °C together with detergent-solubilized HEK293 membrane fractions containing γ-secretase and cleavage efficiency was analyzed as outlined previously[40]. Samples incubated at 4 °C or at 37 °C and in the presence of 0.5 μM of the γ-secretase specific inhibitor L-685,458 (Merck Millipore)[58] served as controls (Supplementary Fig 8). The in vitro generated cleavage products (AICD and Aβ) were analyzed by immunoblotting using antibody 2D8 and Y188, respectively, and quantified by measuring the chemiluminescence signal intensities with the LAS-4000 image reader (Fujifilm Life Science, USA). The signal intensities were quantified with Multi Gauge V3.0 software(Fujifilm Life Sciences). These values were used to generate the plotted ratios (product/(product + precursor)). Quantification of cleavage products for each of the tested substrates was obtained from five independent assays (N = 5). Measurements were taken from distinct samples.

**Mass spectrometric analysis of substrate cleavage products**. For analysis of the different AICD and Aβ species produced by γ-secretase mass spectrometry analysis was performed as previously described[41]. In brief, after incubation of purified substrate with detergent-solubilized membrane fractions containing γ-secretase, an immunoprecipitation step using the antibodies Y188 (AICD) and 4G8 (Aβ) was performed. Samples were diluted using IP-MS buffer [0.1% N-octyl glucoside, 140 mM NaCl, and 10 mM Tris (pH 8.0)] and incubated with the Y188 antibody and Protein A-Sepharose or with 4G8 and Protein G-Sepharose overnight at 4 °C. The immunoprecipitates were washed three times with IP-MS buffer and distilled water and finally eluted (0.1% trifluoroacetic acid in 50% acetonitrile, saturated with a-cyano-4-hydroxy cinnamic acid). Subsequently, samples were subjected to mass spectrometry analysis on a 4800 MALDI-TOF/TOF Analyzer (Applied Biosystems/MDS SCIEX).

**Statistics and reproducibility—mass spectrometry**. Residue-specific DHX kinetics (Eqs. (2) and (3)) originate from time-dependent deuteron contents $D_{mean}$ averaged from the masses of different fragment ions. The residue-specific $D_{mean}$ values were obtained from raw MS data after applying a smoothing function to the D contents of series of c- and z-fragments produced by ETD at different time points. As a result of the smoothing procedure, the precision of the deuteron contents is significantly improved. Rate constants $k_{exp, DHX}$ were determined by a non-linear least squares fitting routine. Standard errors of $\log k_{exp,DHX}$ result from the errors of the fits. The limits of the standard confidence interval of $\Delta G$ are calculated by means of the standard error of $\log k_{exp,DHX}$. A detailed account of the procedure is presented in Supplementary Note 2 as well as in the manual associated with the computer code at Doi: 10.5281/zenodo.7223537.

For mass spectrometric analysis of AICD and Aβ generated in the in vitro cleavage assays, the cleavage products were immunoprecipitated and subsequently analyzed via MALDI-TOF mass spectrometry. One such assay represents one independent experiment. The AICD and Aβ spectra were generated from separate assays. The AICD spectra were measured from three (N = 3) and the Aβ spectra from two independent experiments (N = 2). In both cases, representative MALDI-TOF spectra are shown.

**Statistics and reproducibility—cleavage assays**. For one in vitro cleavage assay, each of the constructs analyzed in this study was incubated together with detergent-solubilized γ-secretase. One such assay represents one independent experiment. Cleavage efficiency was analyzed for each assay separately and quantifications were done from five independent cleavage assays (N = 5). Statistical significance was assessed using one-way ANOVA with Dunnett's multiple comparison test (ns: p > 0.05; *p < 0.05, **p < 0.01, ***p < 0.001, ****p < 0.0001). Two group comparisons were done using pL (Fig. 3) or pL-VGGV (Figs. 4 and 5) as a control group for a multiple comparisons against all other constructs. Data are represented as means ± SEM. For in cellulo assays, the cleavage efficiencies of the constructs were tested in five independent experiments (N = 5), with the exception of pL-VGGV/cr construct that was quantified in four independent experiments (N = 4). Statistical significance was assessed using one-way ANOVA with Dunnett's multiple comparison test (ns: p > 0.05; *p < 0.05, **p < 0.01, ***p < 0.001, ****p < 0.0001). One group comparison was done using pL as a control group for a multiple comparison against all other constructs. Data are represented as means ± SEM.

**In cellulo cleavage assays**. The HEK293 cells were cultured in DMEM media (Gibco) supplemented with 10% FCS in BioCoat™ Poly-D-Lysine 24-well (Corning) plates. The constructs were generated as mentioned in the section on C99 substrate constructs. Transfection of the indicated constructs was performed by Lipofectamine™ 2000 (Invitrogen), according to the manufacturer's instructions. After overnight incubation with transfection solution, following 1× PBS wash, 500 μL of fresh media containing 1 μM DAPT (or vehicle DMSO) were added to the transfected cells. Conditioned media and cell lysates were collected after 24 h incubation. The conditioned media were centrifuged (1 h, 180,000 × g, 4 °C) to pellet cell debris and exosomes. Immunoblotting was performed to measure the Aβ release into the conditioned media from the transiently transfected HEK293 cells. Cell lysates were prepared with STET lysis buffer (150 mM NaCl, 50 mM Tris pH 7.5, 2 mM EDTA, 1% Triton X-100) with protease inhibitors (Roche)[59]. For protein analysis by immunoblotting, samples were prepared as described[59]. Measurements were taken from distinct samples. Following antibodies were used in the immunoblots in given dilutions, anti-HA.11 (1:1000) (Covance, MMS-101P), anti-FLAG (1:2000) (Sigma-Aldrich, F1804), anti-Calnexin (1:2000) (Enzo, ADI-SPA-860).

**Flow cytometry**. HEK293 cells were seeded in 12-well plates. The cells were transfected as described above. After overnight incubation with transfection solution, following 1× PBS wash, 500 μL of fresh media was added. 48 hours after transfection, the cells were suspended with EDTA treatment, and incubated with Alexa Fluor 488 conjugated HA.11 antibody (BioLegend 901509), which targets the extracellular domain of the constructs, or an isotype-matching control antibody (BioLegend 400129). The stained cells, and the control cells, were measured by flow cytometry with a bandpass filter of (527 ± 32) nm (BD FACS Melody), and analyzed by FlowJo v10.4.1 software (BD Life Sciences). For each sample, 100,000 events were acquired and debris, cell aggregates, as well as dead cells, were excluded based on forward and side scatter plots and Propidium iodide (Sigma) staining respectively.

**Biological material availability**. Plasmids encoding recombinant proteins can be obtained from the authors upon request.

**Reporting summary**. Further information on research design is available in the Nature Portfolio Reporting Summary linked to this article.

## Data availability
Source data obtained by ESI mass spectrometry (files 'Kinetik_*') as well as by immunoblotting (file 'Source_data_immunoblotting') are provided at Mendeley Data, V1, Doi: 10.17632/rg8pj45mkc.1 ("Cooperation of N- and C-terminal substrate transmembrane domain segments in intramembrane proteolysis by γ-secretase"). MALDI data are shown in the paper.

## Code availability
The computer code associated with evaluating DHX data in this study has been deposited at Doi: 10.5281/zenodo.7223537.

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

## Acknowledgements

We thank Dr. Axel Imhof for providing access to the mass spectrometers within the Protein Analysis Unit of the LMU, Dr. Christina Scharnagl for helpful discussions, Manuel Hitzenberger for a script to analyze NMR structures, and Fabian Schmidt and Martin Ortner for critically reading the manuscript. This work was funded by the Deutsche Forschungsgemeinschaft (DFG, German Research Foundation) 263531414/ FOR 2290 (H.S., S.L., and D.L.) and the Munich Cluster for Systems Neurology (EXC 2145 SyNergy - ID 390857198). Nadine Werner was funded by a stipend from the Hans and Ilse Breuer Foundation.

## Author contributions

N.W. performed in vitro cleavage assays, P.H. and W.S. did D.H.X. assays, G.G. and M.A. performed in cellulo cleavage assays, M.W. designed software for the analysis and interpretation of DHX mass spectrometry. D.L., S.L., and H.S. designed and supervised the project. D.L. wrote the manuscript together with the co-authors. All authors analyzed their data, created their figures, and have given approval to the final version of the manuscript.

## Funding

## Competing interests

The authors declare no competing interests.

## Additional information

**Peer review information** : *Communications Biology* thanks Oliver Crook and the other, anonymous, reviewer(s) for their contribution to the peer review of this work. Primary Handling Editors: Mark Collins and Gene Chong. Peer reviewer reports are available.

