## [Peer Review File · Communications Biology]

Reviewers' comments:

Reviewer #1 (Remarks to the Author):

Werner and their co-authors present a study on intramembrane proteases which are challenging to study because the cleavage process occurs natively within the lipid bilayer. This study focuses particularly on DHX-MS and cleavage assays, following similar strategies to their previous work. I am not an expert on protease function and so I cannot assess the suitability of these assays to their questions at hand. The manuscript was nicely written and clear to follow. I think the authors could clarify a few points and reformulate some of the figures. A small amount of additional work is needed to fully convince me the mass-spectrometry analysis is sound, but I believe this should not be challenging for the authors.

Major points:

1) It is not clear exactly how the authors determined when the biexponential model was better and if this was statistically rigorous. I struggled to completely follow the details of the statistics in the supplementary, for example how the standard deviations were calculated.

a) Given that it's quite important to their study, the analysis of whether a biexponential model vs mono-exponential model fits better needs to be performed carefully. For example, given that the biexponential model has more parameters and hence will always fit the model better than the mono exponential. The authors should therefore appeal to statistical significance. The correct way to do this is to fit both models to the data, compute the log-likelihood and appeal to Wilk's theorem for a p-value. I think this would strengthen the paper.

2) The experimental design and premise is quite difficult to follow without referring to the previous manuscripts of these authors. To make the paper self-contained and accessible to a wide audience, I would suggest they make an introductory figure covering the experimental design.

3) Could the authors plot representative spectra for the bimodal situations, in the supplementary material?

4) How are the authors sure that these are really EX1 kinetics, did they check for carry-over or other experimental possibilities? I would expect the experimental design to have included blanks.

5) In general it is not clear how the quality of the MS/MS spectra were evaluated and what filtering was applied. Is this automatic within the software?

6) Figure 1 is quite difficult to follow and is quite information dense.

a) Continuity is implied by the connecting panels but the residues are not connected by indication of the residue numbers above.

b) It's essentially impossible to make any inferences about the size of the points based on populations A and B from the figure as there is too much other information. I would place this information in another figure

c) It is not clear what A and B refer to without going to the methods, which seems unnecessary.

d) The time-axis are all different, were the data not obtained consistently?

e) Fitted lines extend beyond the data.

f) Many other figures suffer from the same issue.

7) The mechanisms of substrate selection in the discussion are interesting. If possible, would a cartoon help clarify the author's proposals?

8) Code and data need to be placed in the appropriate repository. Code should be placed in a zenodo repository.

Minor points:

1) The authors refer to "standard confidence intervals", could they be more precise? In particular, what is the size (95%)?

2) Perhaps I missed it, but it was not clear how quantification of the immunoblotting data was performed.

3) Figure 2 C and E, the axis extend beyond 100%

Reviewer #2 (Remarks to the Author):

Werner et al. address a long-standing question in the protease field of how the intramembrane protease presenilin/gamma-secretase selects its substrates while the majority of transmembrane domains (TMDs) are not cleaved. Although this question has already been approached by several previous studies, including reports from the authors, the underlying principles are still mysterious. The importance of a TMD hinge region of the amyloid precursor protein (APP)-derived substrate known as C99, and whether helix-unfolding around the cleavage site plays an important discriminatory role, are still not fully resolved. In this study, the authors combine high-resolution deuterium/hydrogen exchange (DHX) experiments, *in vitro* gamma-secretase cleavage assays with cell-based studies. Although the biophysical approach builds on a compromise, the use of a trifluoroethanol/water mix as a mimic for the aqueous environment within the presenilin active site, the data is of extreme high quality and all conclusions are justified. First, using a cutting-edge DHX protocol, the authors compare four well-known model gamma-secretase substrates, thereby revealing TMD "helix fraying" as a general substrate pattern. Interestingly, the authors observe double H-bond openings as a hallmark of TMD helix fraying. Building on their comparative substrate TMD analysis and previous studies, the authors isolate two substrate features in APP-C99, namely a di-glycine hinge and the C-terminal cleavage site region, and tested their influence on the dynamics and processing of a poly-leucine stretch. This systematic and rigorous analysis revealed that N-terminal TMD hinge conferred partial cleavability whereas flexibility within the natural C-terminal cleavage region appeared not to be crucial. Despite that, for full cleavability of C99 both features were required, suggesting that N- and C-terminal segments covertly enable gamma-secretase-catalyzed cleavage. All experiments are carefully conducted, data is of high quality including appropriate controls and statistical tests. Only minor concern is that in by overexpressing their model substrates in the cell-based gamma-secretase assay the authors need to confirm (e.g. by immunofluorescence microscopy) that the uncleaved poly-leucine non-substrate reaches the late secretory pathway, where the active gamma-secretase is located.

Rebuttal letter - COMMSBIO-22-0909-T

Reviewer #1 (Remarks to the Author):

Werner and their co-authors present a study on intramembrane proteases which are challenging to study because the cleavage process occurs natively within the lipid bilayer. This study focuses particularly on DHX-MS and cleavage assays, following similar strategies to their previous work. I am not an expert on protease function and so I cannot assess the suitability of these assays to their questions at hand. The manuscript was nicely written and clear to follow. I think the authors could clarify a few points and reformulate some of the figures. A small amount of additional work is needed to fully convince me the mass-spectrometry analysis is sound, but I believe this should not be challenging for the authors.

We thank the reviewer for the careful perusal of our manuscript and the valuable comments. Before going into the details, we would like to mention that, we also reviewed the mechanism proposed to underly the biphasic nature of DHX, a technical sideline of this work. In doing so, we found that single exchange events occuring from doubly open states, as proposed in the original version of the manuscript, are unlikely to account for our observed biphasic kinetics for the following reason: Doubly and singly open states must be expected in equilibrium at the time scale of the experiment. In this model, therefore, exchanges of amide deuterons from single or doubly open states would have to be treated as parallel reactions where both rate constants in the end merge to a single rate constant. That is, that previous model is inconsistent with the data.

Literature research suggested another model to account for biphasic DHX. In addition to the prevalent EX2 mode of non-correlated DHX in a helix, this new model assumes that very unstable regions of our TMD helices, such as a glycine-rich hinge or frayed helix termini, may undergo occasional correlated exchange (EX1 mode) that is limited to two (or even more) neighboring deuterons in addition to non-correlated DHX. The isotope patterns of both modes would overlap, thus shifting the average masses of the combined patterns to lower values (assuming EX1 exchange of 1 D and overlapping EX2 exchange of 2 D or more). As a result, a mixed EX1/EX2 regime may mimic accelerated initial correlated exchange, as detailed in the revised Supplementary Discussion. In sum, this new model is well suited to explain fast initial kinetics followed by slower kinetics at flexible helix regions. The slow kinetics allow us to calculate thermodynamic stabilities of given amide bonds. At the same time, the apparent existence of correlated exchange events is a valuable additional hint at local helix flexibility. We therefore report H-bond stabilities based on the slow part of the kinetics and indicate regions of biphasic exchange kinetics with shading in the revised figures. The previous model in Fig. S2 is replaced by the new model.

Importantly, this modified interpretation of our DHX results does in no way alter our conclusions on the relevance of local helix flexibility for the mechanism by which by gamma-secretase selects

its substrate TMDs, which is the crucial point of this work. Related changes in the text are in green while changes in response to the reviewer's comments are in red. The updated Figures within the main text are given at the end of this response.

Major points:

1) It is not clear exactly how the authors determined when the biexponential model was better and if this was statistically rigorous. I struggled to completely follow the details of the statistics in the supplementary, for example how the standard deviations were calculated. a) Given that it's quite important to their study, the analysis of whether a biexponential model vs mono-exponential model fits better needs to be performed carefully. For example, given that the biexponential model has more parameters and hence will always fit the model better than the mono exponential. The authors should therefore appeal to statistical significance. The correct way to do this is to fit both models to the data, compute the log-likelihood and appeal to Wilk's theorem for a p-value. I think this would strengthen the paper.

This is a very good point, we have now tested fitting our kinetics data with different p-values using Wilks' theorem using a correspondingly expanded analysis software. We identified a p-value of 0.01 as a sensible threshold between monoexponential and biexponential fitting. Using this threshold, we have reevaluated all of our DHX data in order to treat them in a comparable fashion. In most cases, the results were very similar to the outcomes in the original manuscript. The errors of amide exchange data are given using confidence intervals which are equivalent to the standard errors (see below). Standard deviations were not calculated here since we considered the variance of the data to be less important than the significance of any difference found between TMD constructs.

2) The experimental design and premise is quite difficult to follow without referring to the previous manuscripts of these authors. To make the paper self-contained and accessible to a wide audience, I would suggest they make an introductory figure covering the experimental design.

Good idea! To make the paper more accessible, we made an introductory Scheme 1 covering the experimental design.

Scheme 1

3) Could the authors plot representative spectra for the bimodal situations, in the supplementary material?

Well, the point is that the raw spectra after gas phase fragmentation are extremely complex since they contain dozens to hundreds of c- and z-fragments of different masses, charge states and deuteration grades. Therefore, looking at the raw data does not provide much information on the process, i.e. monomodal vs bimodal DHX, that has generated them. Rather, deciding whether they represent biphasic or monophasic DHX is only possible after careful interpretation using the described procedure.

4) How are the authors sure that these are really EX1 kinetics, did they check for carry-over or other experimental possibilities? I would expect the experimental design to have included blanks.

I suppose, the reviewer really refers to EX2 (rather than EX1). For the C99 wild-type transmembrane helix we have previously (Pester, O. et al. The Backbone Dynamics of the

Amyloid Precursor Protein Transmembrane Helix Provides a Rationale for the Sequential Cleavage Mechanism of γ -Secretase. J. Am. Chem. Soc. 135, 1317-1329 (2013)) shown that the isotope envelope gradually shifts with incubation time towards lower mass/charge values due to gradual DHX. This is cited now in the Supplementary Discussion of the submitted manuscript. A gradual mass shift is diagnostic of uncorrelated exchange, i.e. EX2 mechanism. By contrast, pure EX1 kinetics would be characterized by gradual diminution of the peak intensity of the deuterated species paralleled by gradual increase of the protonated species. As noted above, we now suggest that occasional and limited EX1 DHX may occur in parallel to the prevalent EX2 mode to explain the biphasic nature of exchange in case of very flexible regions of our helices.

5) In general it is not clear how the quality of the MS/MS spectra were evaluated and what filtering was applied. Is this automatic within the software?

As detailed in the Statistics section (lines 525-528), we used raw MS/MS data without prior filtering. Specifically, the residue-specific D_{mean} values were obtained from raw MS data after applying a smoothing function to the D contents of series of c- and z-fragments produced by ETD at the different time points. As a result of the smoothing procedure, the precision of the deuterium contents is significantly improved.

6) Figure 1 is quite difficult to follow and is quite information dense.
a) Continuity is implied by the connecting panels but the residues are not connected by indication of the residue numbers above.

Sure, we have now separated the panels to make it clear that these sequences are not connected and represent entirely different peptides.

b) It's essentially impossible to make any inferences about the size of the points based on populations A and B from the figure as there is too much other information. I would place this information in another figure

We have transferred the information on population sizes to Supplementary Table S2 where it is listed together with kinetic constants.

c) It is not clear what A and B refer to without going to the methods, which seems unnecessary.

A statement on the meaning of population sizes A and B is now made in the Supplementary Discussion.

d) The time-axis are all different, were the data not obtained consistently?

This was done on purpose. To make this clear, the legend to Fig. 1 says of the lower panels shown in part B: "linear time axes were truncated such as to better visualize the intersections between fast and slow regimes of exchange (arrows)" (lines 138-139). In other words, data shown by lower panels represent part of the data shown in upper panels.

e) Fitted lines extend beyond the data.

Fitted lines beyond the data have been removed.

f) Many other figures suffer from the same issue.

We hope that we have taken care of all instances suffering from those issues.

7) The mechanisms of substrate selection in the discussion are interesting. If possible, would a cartoon help clarify the author's proposals?

We now made a cartoon (Scheme 2) to visualize the mechanism proposed by us.

Scheme 2

8) Code and data need to be placed in the appropriate repository. Code should be placed in a zenodo repository.

We have placed raw data in the Mendeley server (doi: 10.17632/rg8pj45mkc.1) and the code into the Zenodo server (doi: 10.5281/zenodo.7223537) as described in the Code Availability Statement given in the manuscript (lines 582-588).

Minor points:

1) The authors refer to “standard confidence intervals”, could they be more precise? In particular, what is the size (95%)?

The confidence intervals are equivalent to the standard errors, i.e. they correspond to the fraction of data covered by one standard deviation, a generally accepted statistical measure.

2) Perhaps I missed it, but it was not clear how quantification of the immunoblotting data was performed.

In addition to the information already supplied in the Methods section, we extend Methods by "The signal intensities were quantified with Multi Gauge V3.0 software(Fujifilm Life Sciences).These values were used to generate the plotted ratios (product/(product + precursor)) (lines 509-511).

3) Figure 2 C and E, the axis extend beyond 100%
We have now corrected this.

Our revised figures:

Fig. 1

Fig. 2

Fig. 4

Reviewer #2 (Remarks to the Author):

Werner et al. address a long-standing question in the protease field of how the intramembrane protease presenilin/gamma-secretase selects its substrates while the majority of transmembrane domains (TMDs) are not cleaved. Although this question has already been approached by several previous studies, including reports from the authors, the underlying principles are still mysterious. The importance of a TMD hinge region of the amyloid precursor protein (APP)-derived substrate known as C99, and whether helix-unfolding around the cleavage site plays an important discriminatory role, are still not fully resolved. In this study, the authors combine high-resolution deuterium/hydrogen exchange (DHX) experiments, in vitro gamma-secretase cleavage assays with cell-based studies. Although the biophysical approach builds on a compromise, the use of a trifluoroethanol/water mix as a mimic for the aqueous environment within the presenilin active site, the data is of extreme high quality and all conclusions are justified. First, using a cutting-edge DHX protocol, the authors compare four well-known model gamma-secretase substrates, thereby revealing TMD “helix fraying” as a general substrate pattern. Interestingly, the authors observe double H-bond openings as a hallmark of TMD helix fraying. Building on their comparative substrate TMD analysis and previous studies, the authors isolate two substrate features in APP-C99, namely a di-glycine hinge and the C-terminal cleavage site region, and tested their influence on the dynamics and processing of a poly-leucine stretch. This systematic and rigorous analysis revealed that N-terminal TMD hinge conferred partial cleavability whereas flexibility within the natural C-terminal cleavage region appeared not to be crucial. Despite that, for full cleavability of C99 both features were required, suggesting that N- and C-terminal segments covertly enable gamma-secretase-catalyzed cleavage. All experiments are carefully conducted, data is of high quality including appropriate controls and statistical tests.

We thank the reviewer for the careful perusal of our manuscript and the valuable comments. Before going into the details, we would like to mention that, we also reviewed the mechanism proposed to underly the biphasic nature of DHX, a technical sideline of this work. In doing so, we found that single exchange events occurring from doubly open states, as proposed in the original version of the manuscript, are unlikely to account for our observed biphasic kinetics for the following reason: Doubly and singly open states must be expected in equilibrium at the time scale of the experiment. In this model, therefore, exchanges of amide deuterons from single or doubly open states would have to be treated as parallel reactions where both rate constants in the end merge to a single rate constant. That is, that previous model is inconsistent with the data.

Literature research suggested another model to account for biphasic DHX. In addition to the prevalent EX2 mode of non-correlated DHX in a helix, this new model assumes that very unstable regions of our TMD helices, such as a glycine-rich hinge or frayed helix termini, may undergo occasional correlated exchange (EX1 mode) that is limited to two (or even more) neighboring deuterons in addition to non-correlated DHX. The isotope patterns of both modes would overlap, thus shifting the average masses of the combined patterns to lower values (assuming EX1 exchange of 1 D and overlapping EX2 exchange of 2 D or more). As a result, a mixed EX1/EX2 regime may mimic accelerated initial correlated exchange, as detailed in the revised

Supplementary Discussion. In sum, this new model is well suited to explain fast initial kinetics followed by slower kinetics at flexible helix regions. The slow kinetics allow us to calculate thermodynamic stabilities of given amide bonds. At the same time, the apparent existence of correlated exchange events is a valuable additional hint at local helix flexibility. We therefore report H-bond stabilities based on the slow part of the kinetics and indicate regions of biphasic exchange kinetics with shading in the revised figures. The previous model in Fig. S2 is replaced by the new model.

Importantly, this modified interpretation of our DHX results does in no way alter our conclusions on the relevance of local helix flexibility for the mechanism by which by gamma-secretase selects its substrate TMDs, which is the crucial point of this work. Related changes in the text are in green while changes in response to the reviewer's comments are in red.

Only minor concern is that in by overexpressing their model substrates in the cell-based gamma-secretase assay the authors need to confirm (e.g. by immunofluorescence microscopy) that the uncleaved poly-leucine non-substrate reaches the late secretory pathway, where the active gamma-secretase is located.

Good point! We did a control flow cytometric experiment to adress this issue. The experiment revealed that, compared to wt C99, the pL construct is slightly less, but still efficiently expressed at the cell surface (shown in Fig. S7), thus ruling out intracellular retention of pL as a potential source of its near-complete resistance to cleavage.

Figure S7. Comparing cell surface localization of wt C99 and its pL derivative. (A) HEK293E cells were transfected with the respective constructs. The treated cells were suspended and labeled with HA antibody that targets the extracellular site of the constructs, or isotype control (ctrl). Shown are representative histograms from N=5 experiments. (B) The relative surface abundance of the measurement from panel A. The geometric mean of the HA-signal was divided by the respective

geometric mean of the Ig-signal to obtain the mean fluorescent intensity. The C99 surface signal was used as baseline, and its average normalized to 1. A two-tailed unpaired t-test was used. (C) Cells measured in panel A were collected, lysed and blotted against the HA-tag or calnexin as loading control. Shown are representative blots from N=4 experiments. (D) Quantification of blots in panel C. The C99 protein expression was used as baseline, and its average normalized to 1. A two-tailed unpaired t-test was used.

REVIEWERS' COMMENTS:

Reviewer #1 (Remarks to the Author):

I believe the manuscript is much improved through the revision and they have addressed my comments thoroughly. I have no additional comments about the manuscript. It is a very piece of work.

Reviewer #2 (Remarks to the Author):

The authors have convincingly address my initial concern.